# Molecular characterisation of *Trypanosoma cruzi* in *Triatoma dimidiata* from a highland locality in western Panamá

Vanessa Jenny Pineda[1], Kadir González[1,2], José Eduardo Calzada[1,3]/+, Azael Saldaña[4]/+

[1]Instituto Conmemorativo Gorgas de Estudios de la Salud, Departamento de Investigaciones en Parasitología, Panamá, Panamá
[2]Universidad de Panamá, Facultad de Medicina, Departamento de Microbiología Humana, Panamá, Panamá
[3]Universidad de Panamá, Facultad de Medicina Veterinaria, Panamá, Panamá
[4]Universidad de Panamá, Facultad de Medicina, Centro de Investigación y Diagnóstico de Enfermedades Parasitarias, Panamá, Panamá

**BACKGROUND** *Triatoma dimidiata* is a widely distributed vector of *Trypanosoma cruzi* in Mesoamerica, but its epidemiological role in most regions of Panamá remains poorly understood.

**OBJECTIVES** To investigate the presence, infection status, and feeding behaviour of *T. dimidiata* populations in peridomestic areas of Palmira Arriba, western Panamá.

**METHODS** Entomological surveys were conducted in five peridomestic sites of a rural highland community. Thirty-seven triatomines (13 adults and 24 nymphs) were collected from wooden piles and construction materials in contact with the ground. DNA from 30 specimens was analysed by polymerase chain reaction (PCR) for *T. cruzi* detection, genotyping [discrete typing unit (DTU) and haplotype identification], and blood meal source determination through cytochrome b amplification.

**FINDINGS** Twenty-one insects (70.0%) were positive for *T. cruzi*. Sixteen infections (76.2%) belonged to DTU I (TcI), including 13 TcIDOM and 14 TcIa genotypes, both linked to domestic and sylvatic cycles. Blood meal analysis revealed one mammalian and two avian feedings, indicating opportunistic behaviour.

**MAIN CONCLUSIONS** This study provides the first molecular confirmation of *T. cruzi* infection in *T. dimidiata* from Palmira Arriba. The combination of high infection prevalence, multiple developmental stages, and recent feeding suggests active local transmission favoured by humid and cool ecological conditions. Expanded surveillance and integrative One Health approaches are needed to elucidate transmission dynamics in highland rural Panamá.

Key words: *Trypanosoma cruzi* - discrete typing unit - *Triatoma dimidiata* - molecular epidemiology - Panamá

Chagas disease (CD), caused by the protozoan parasite *Trypanosoma cruzi*, remains a significant vector-borne zoonosis in Latin America, with over seven million people infected worldwide.[1] In Panamá, *T. cruzi* transmission is primarily linked to sylvatic and peridomestic cycles involving triatomine vectors and mammalian reservoirs.[2] Historically, *Rhodnius pallescens* has been regarded as the principal vector due to its strong ecological association with the endemic palm tree *Attalea butyracea* and its widespread presence across central Panamá.[3] However, the increasing detection of *Triatoma dimidiata* infestations and evidence of its colonisation in rural dwellings of western provinces, including Veraguas and Chiriquí, suggest a shifting entomological landscape that merits further investigation.[4,5]

*Triatoma dimidiata* is a widely distributed triatomine species that ranges from southern México to northern South America, including Colombia, Ecuador, and parts of Venezuela.[6,7] While often considered a secondary vector in Panamá, *T. dimidiata* has demonstrated remarkable ecological plasticity, thriving in sylvatic, peridomestic, and domestic environments.[8,9,10] Its ability to invade and colonise rural households, combined with a broad host feeding range that includes humans, dogs, chickens, and synanthropic mammals, confers a high potential for *T. cruzi* transmission.[11,12,13] Despite this, molecular surveillance data on *T. dimidiata* populations and their infection dynamics with *T. cruzi* remain scarce in Panamá, particularly in highland regions such as Chiriquí province, where ecological conditions may favour persistent vector populations.

Historical evidence of *T. dimidiata* presence in the Chiriquí province dates back several years. Entomological surveys conducted in 1986, 1999, and 2012 documented the collection of triatomine bugs in the corregimiento of Palmira, Boquete district.[5] A total of 131

Financial support: This study was carried out with funding from the Ministry of Economy and Finance (MEF) of the Republic of Panamá, Project No. 0420. Additional support was provided by the National Research System (SNI) of Panamá through research incentives awarded to AS, KG and JEC as members of SNI-SENACYT Panamá.
VJP and KG contributed equally to this work.
+ Corresponding authors: jecalzada@gorgas.gob.pa | ⊕ https://orcid.org/0000-0002-6669-4290 /
azael.saldana@up.ac.pa | ⊕ https://orcid.org/0000-0002-5653-1332

**Handling editor:** Adeilton Alves Brandão | ⊕ https://orcid.org/0000-0001-5877-607X

specimens were captured during that period; however, microscopic examination of the insects yielded negative results for *T. cruzi*. Additionally, preliminary serological screening using a latex agglutination test among 100 residents revealed a 2% positivity rate. Although constrained by the diagnostic tools available at the time, these findings raised concerns about the possibility of undetected *T. cruzi* transmission cycles and underscored the need for further investigation.

In support of ongoing surveillance efforts, the Department of Parasitology Research at the Gorgas Memorial Institute received 30 *T. dimidiata* specimens from the same community of Palmira Arriba over the past decade. These samples were processed as part of a routine diagnostic service for species identification and *T. cruzi* infection assessment, but the data were not previously published. All were independent of the present field collections. All specimens were taxonomically confirmed as *T. dimidiata* and collected from peridomestic ecotopes. Microscopic examination revealed that 16.7% (5/30) of the triatomines were infected with *T. cruzi*.

Palmira Arriba, situated in the highland region of Boquete, constitutes an ecologically distinct area that, until now, had not been the subject of molecular studies concerning triatomine-borne *T. cruzi* transmission. In this context, we conducted a preliminary molecular study aimed at determining the infection prevalence of *T. cruzi* in *T. dimidiata* specimens collected from Palmira Arriba and characterising the genetic diversity of the circulating parasite.

The objective of this pilot investigation was to elucidate the eco-epidemiological role of *T. dimidiata* in this newly recognised transmission focus by integrating field entomological surveys with molecular detection and typing methods. The findings contribute to a more comprehensive understanding of CD transmission dynamics in western Panamá and underscore the importance of updating national vector surveillance and control strategies considering emerging local transmission patterns.

## SUBJECTS AND METHODS

*Study site and Triatomine collection procedures* - Palmira Arriba (8.76716º N, 82.45815º W) is a small rural community located in the highlands of Boquete District, Chiriquí Province, Panamá, approximately 478 kilometres (a 7-hour drive) from Panamá City (Fig. 1).[5,14] The area has an average elevation of 1,300 metres above sea level and is characterised by a cool, humid montane tropical climate, with annual rainfall reaching up to 3,100 mm and temperatures ranging from 17ºC to 28ºC.[15] The population is estimated at around 300 inhabitants. Local livelihoods are based primarily on small scale agriculture (including coffee, vegetables, and fruits), as well as ecotourism.[16] The surrounding landscape is dominated by cloud forest, supporting high biodiversity and a complex ecological interface between humans, domestic animals, and wildlife.[17]

Palmira Arriba was chosen as the study site due to its long-recognised presence of *T. dimidiata* populations, documented in previous entomological surveys conducted in 1986, 1999, and 2012.[5] During those surveys, a small

serological screening of residents revealed 2% positivity rate for *T. cruzi* antibodies, suggesting possible low-level or undetected transmission cycles. Despite these early observations, no molecular investigations of vectors or parasites had ever been performed in this community prior to the present study. This historical background, together with its distinctive highland ecology, justified the selection of Palmira Arriba as a priority site for molecular and eco-epidemiological evaluation of *T. cruzi* transmission.

Triatomine surveys were carried out in collaboration with staff from the Vector Control Department of the Ministry of Health (MINSA), Chiriquí Province. Manual searches were conducted in the peridomestic areas of five households, following prior informed consent from property owners. Five distinct peridomestic household areas were surveyed, located within 0.5-1.5 km of each other. These included piles of timber and wooden materials, animal enclosures (dog/cat resting areas and chicken coops), and adjacent storage structures within household premises. A range of potential triatomine habitat patches was inspected, including piles of construction timber, wooden materials, animal enclosures (such as dog/cat bedding areas and chicken coops), and various storage structures.

Each triatomine specimen collected was placed in a separate plastic container, labelled, and transported to the laboratory for processing. Taxonomic identification was performed to confirm species and developmental stage. Adult specimens were identified following morphological keys.[18,19] For nymphal stages, identification was based on the rearing of representative individuals to adulthood in the laboratory and corroboration of distinctive morphological traits observable in late instars. This limitation is acknowledged as a potential source of uncertainty in nymphal identification. Dissections were conducted under a stereomicroscope to extract the intestinal tract, which was stored at -20ºC until DNA analysis.

Molecular detection and genotyping of *T. cruzi* - Genomic DNA was extracted from the intestinal material using the Wizard® Genomic DNA Purification Kit (Promega), following the manufacturer's instructions. Detection of *T. cruzi* was performed using conventional polymerase chain reaction (PCR) with the primers S35/S36, which amplify a 330 bp fragment of the variable region of *T. cruzi* minicircle DNA.[20]

Positive samples were subsequently analysed by a real-time PCR for discrete typing unit (DTU) assignment, using a panel of molecular markers targeting SL-IR (TcI–TcIII), COII (TcII–TcIV), ND1 (TcV), and 18S rRNA (TcVI), following the protocol described by Muñoz-San Martín et al.[21] For samples identified as TcI, subgenotyping was conducted to discriminate between TcIDOM and sylvatic TcI strains, using the SL-IR region as a molecular marker.[22] Further subtyping to distinguish TcIa, TcIb, and TcId haplotypes was carried out using primers described by Falla et al.[23]

*Detection of blood meal sources* - To identify the blood meal source, DNA was extracted from the intestinal contents of triatomines and subjected to PCR amplification targeting the mitochondrial cytochrome b (Cyt b)

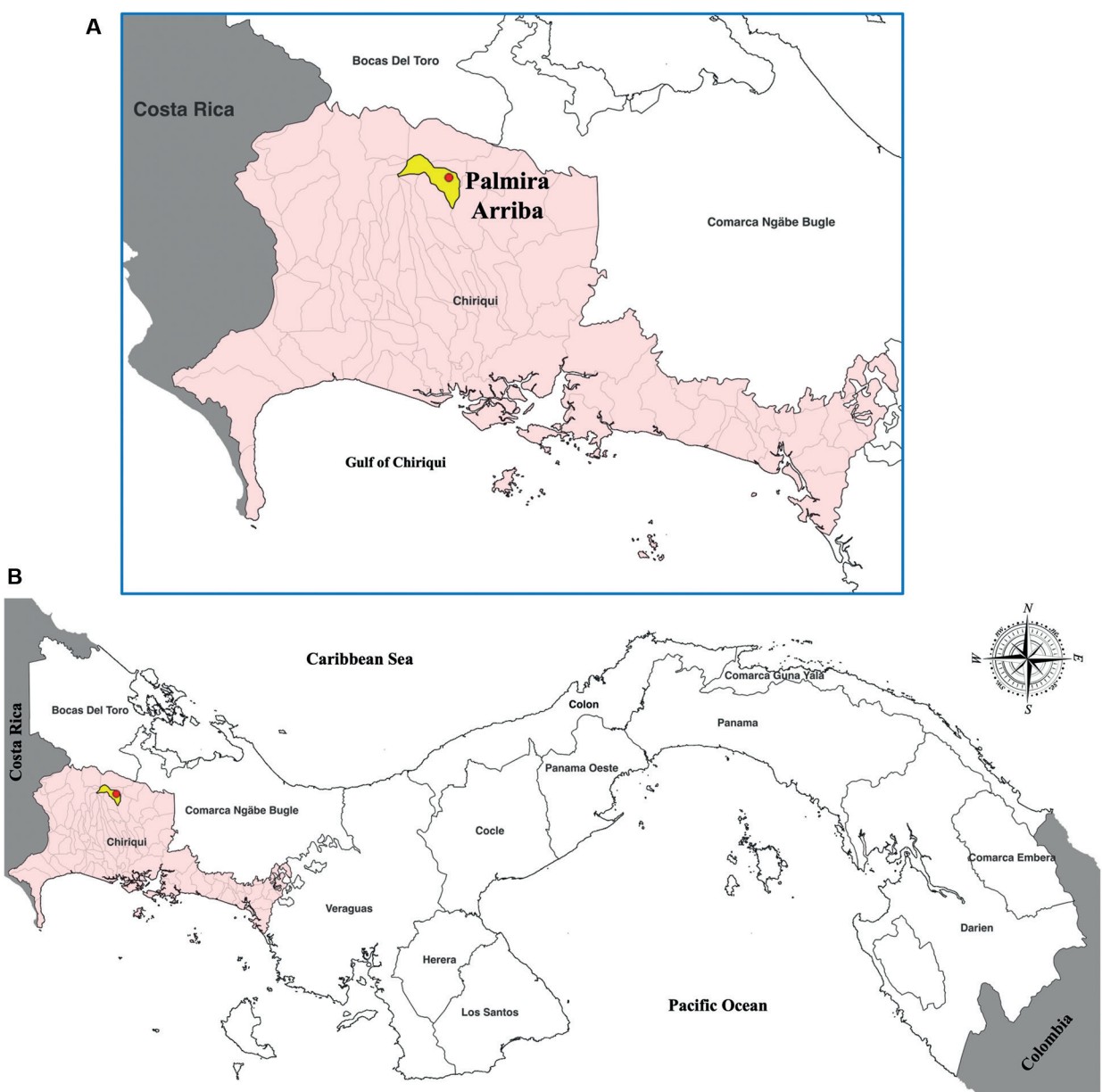

Fig. 1: map of Panamá showing the geographic location of the study area within the Chiriquí Province. (A) map shows Boquete District (highlighted in yellow) in Chiriquí Province, western Panamá. The corregimiento of Palmira, where the study was conducted, is highlighted in red. (B) map displays the location of Chiriquí Province within the national territory of Panamá.

gene. Two primer sets specific for mammals (772 bp) and birds (508 bp), as described by Ngo and Kramer,[24] were used in separate reactions. PCR products were resolved by electrophoresis on 1.5% agarose gels and visualised using SYBR™ Green I Nucleic Acid Gel Stain.

*Ethical statement* - This preliminary study did not involve human participants or the handling of vertebrate animals. All triatomine collections were conducted exclusively in peridomestic settings, with the prior consent of household owners, and following current environmental and public health guidelines to minimise disruption to local ecosystems and domestic animal habitats. This study was granted exemption (009/CIUCAL/ICGES-2024) from the Comité Institucional para el Uso y Cuidado de Animales de Laboratorio (CIUCAL-ICGES).

## RESULTS

Five peridomestic areas in Palmira Arriba were surveyed, allowing the identification and evaluation of various habitat patches suitable for *T. dimidiata*. A total of 37 triatomines were collected, including 13 (35.1%) adults and 24 (64.9%) nymphs at different developmental stages. All specimens were taxonomically confirmed as *T. dimidiata* and collected from peridomestic ecotopes. Most specimens were associated with piles of construction timber and wooden materials in direct contact with the ground. These structures frequently served as resting or sheltering sites for domestic animals, including dogs, cats, and backyard poultry. Microscopic examination revealed that 16.7% (5/30) of the triatomines were infected with *T. cruzi*. Of the 30 *T. dimidiata* specimens analysed

by PCR, 70% (21/30) tested positive for *T. cruzi*, with a higher infection rate observed in nymphs (71.4%) compared to adults (28.6%). Genotyping revealed that 76.2% (16/21) of the positive samples corresponded to the TcI lineage, predominantly the TcIDOM genotype and TcIa haplotype. The remaining 23.8% of positive samples could not be genotyped, likely due to low DNA quantity or degraded templates, a common limitation in field-collected triatomine specimens. Full genotyping details are provided in Fig. 2 and Table.

The analysis of blood meal sources using Cyt b primers revealed that one triatomines tested positive for mammalian blood and two for avian blood. Most triatomines analysed were starved and lacked recent blood meals, which likely limited PCR detection of vertebrate DNA, as observed in previous studies.[25]

## DISCUSSION

This study provides the first molecular confirmation of *T. cruzi* infection in *T. dimidiata* from Palmira Arriba, a highland rural locality in Chiriquí Province. The infection rate observed (70.0%), one of the highest documented for *T. dimidiata* in the Meso-Andean region, was detected in both adult and nymphal stages, indicating active transmission within peridomestic environments. Comparable or lower infection rates have been reported

in other endemic localities, such as Santa Fé, Veraguas/Panamá (21.4%),[4] northern Belize (60%),[26] Yucatán, México (22.8%),[9] and eastern Colombia (70%).[27]

All positive samples were exclusively associated with DTU I (TcI), consistent with the predominant DTU reported throughout Central America and northern South America.[7,28,29] Further molecular characterisation identified two TcI genotypes: the TcIa haplotype and the TcIDOM genotype, both of which are frequently associated with domestic and sylvatic transmission cycles in the Meso-Andean regio.[22,23,30]

The presence of TcIDOM, a genotype strongly associated with domestic transmission cycles and human infections,[22,30,31] suggests the establishment of a stable peridomestic transmission cycle in Palmira Arriba. This is further supported by the detection of multiple developmental stages of triatomines in piles of construction timber and wooden materials adjacent to homes, structures that likely serve as refuges and breeding sites for the vector, while also functioning as resting or sheltering areas for domestic animals. The finding aligns with studies from other endemic regions, such as Santa Fé, Panamá,[4] Yucatán, México[9,32,33] and northern Belize,[26] where *T. dimidiata* has shown strong peridomestic adaptation and colonisation of structures like chicken coops, doghouses, and opossum nests. These microhabitats, of-

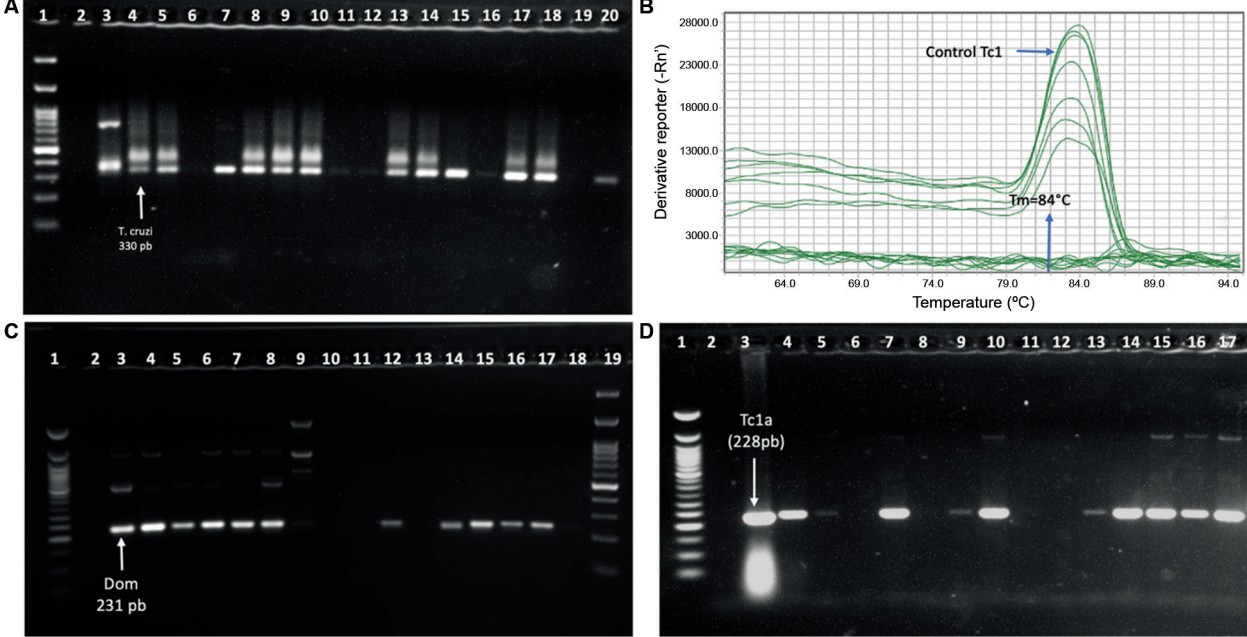

Fig. 2: molecular detection of *Trypanosoma cruzi* in triatomines from Palmira Arriba, Boquete, Panamá. (A) Conventional polymerase chain reaction (PCR) amplification of kinetoplast DNA using S35/S36 primers. Products were resolved on a 1.5% agarose gel stained with SYBR™ Green I Nucleic Acid Gel Stain. Lane 1:100 bp molecular weight marker (Promega); Lane 2: negative control; Lane 3: *T. rangeli* control; Lane 4: *T. cruzi* control; Lanes 5-20: triatomine samples. (B) Real-time PCR using SL-IR primers Tcl SL-IR Fw and Tcl SL-IR Fr in a QuantStudio 5 Real-Time PCR System. Shown: *T. cruzi* TcI control and triatomine samples from Palmira Arriba. (C) Conventional PCR using SL-IR primers 1Am/1B. Products were separated on a 1.5% agarose gel stained with SYBR™ Green I. Lane 1:50 bp molecular weight marker; Lane 2: negative control; Lane 3: *T. cruzi* TcIDOM control; Lanes 4-18: triatomine samples; Lane 19:100 bp molecular weight marker (Promega). (D) PCR amplification of the miniexon intergenic region using primer sets 1A-B (228 bp), 2A-B (250 bp), and 4A-B (200 bp). Products were visualised on a 1.5% agarose gel stained with SYBR™ Green I. Lane 1:50 bp molecular weight marker (Promega); Lane 2: negative control; Lane 3: *T. cruzi* TcIa control; Lanes 4-17: triatomine samples from Palmira Arriba.

TABLE

*Trypanosoma cruzi* infection and genotyping results in *Triatoma dimidiata* (n = 37) collected from Palmira Arriba, Chiriquí, Panamá

| | Number of *T. dimidiata* | Percentage (%) |
|---|---|---|
| Total *T. dimidiata* analysed by PCR | 30 | 100% |
| *T. cruz*i positive | 21 | 70.0% (21/30) |
| - Adults positive | 6 | 28.6% (6/21) |
| - Nymphs positive | 15 | 71.4% (15/21) |
| Genotyped as TcI | 16 | 76.2% (16/21) |
| No amplification | 5 | 23.8 (5/21) |
| - TcIDOM genotype | 13 | 81.2% (13/16) |
| - TcIa haplotype | 14 | 87.5% (14/16) |
| No amplification | 2 | 12.5% (2/16) |

PCR: polymerase chain reaction.

ten referred to as "habitat patches", provide stable food sources and favourable environmental conditions that support the year-round persistence of vector populations and sustained *T. cruzi* transmission.[10,34]

The exclusive detection of TcI in Palmira Arriba is consistent with multiple studies across Panamá that have documented this DTU as the dominant lineage infecting humans, wildlife, and triatomines.[35,36,37] Although only the TcIa haplotype and TcIDOM genotype were identified in this study, broader surveys in México, Colombia, and Belize have revealed mixed infections with other sublineages, such as TcId and TcIV, within individual triatomines.[7,11,38,39]

These findings suggest that the genetic diversity of *T. cruzi* in Palmira Arriba may be broader than currently observed and underscore the need for expanded molecular surveillance to capture the full range of circulating DTUs or haplotypes in the area.

On the other hand, the high infection prevalence observed in triatomines from Palmira Arriba may reflect specific ecological and environmental conditions that facilitate the establishment of triatomine populations and enhance the transmission of *T. cruzi*. Located at approximately 1,300 meters above sea level, Palmira Arriba has a cool and humid tropical mountain climate with frequent rainfall and dense vegetation, conditions similar to those found in other mid to high altitude localities in Chiapas, México,[40] and the Colombian Andes,[6,31] where *T. dimidiata* has demonstrated high vector competence. Notably, experimental evidence from Chiapas suggests that *T. cruzi* isolates from mid elevation zones (∼ 700 m) may exhibit greater virulence and trigger stronger immune responses in both triatomines and mammalian hosts, implying that altitude related ecological factors could influence parasite behaviour and transmission dynamics.[40]

In this ecological context, evidence of host-feeding behaviour further supports the local adaptation of *T. dimidiata* populations. Molecular analysis of blood meal sources revealed that one *T. dimidiata* analysed specimen had fed on mammalian hosts and two on avian hosts, as determined by PCR amplification of cytochrome b gene fragments. Although the low number of positive detections is likely attributable to the analysis of starved specimens lacking recent blood ingestion, the identification of both mammalian and avian blood provides direct evidence of active feeding within the evaluated peridomestic environment. These results are consistent with earlier serological findings from western Panamá,[13] particularly in the district of Boquete, where capillary precipitin tests showed that over 70% of triatomines had fed on mammals, most frequently humans (37.7%) and dogs (17.1%), followed by chickens (19.0%) as the predominant avian source.

This trophic plasticity is not unique to Panamá but has been consistently reported across Mesoamerican and northern South American regions, further illustrating the ecological adaptability of *T. dimidiata*. Molecular studies in México, Central America, and Colombia, have revealed broad host ranges, including humans, dogs, cattle, birds, and wild mammals such as opossums.[12,25,27,39] Dogs frequently emerge as key bridge hosts, and humans are consistently among the most common blood sources. This generalist feeding behaviour likely supports the long-term persistence of vector populations in sites where humans, domestic animals, and sylvatic reservoirs coexist, thereby reinforcing the role of *T. dimidiata* as a competent bridge vector for *T. cruzi* transmission between sylvatic and domestic cycles in the highlands of western Panamá.

From a One Health perspective, the epidemiological role of *T. dimidiata* in this highland area must be reevaluated. Genetic differentiation in *T. dimidiata* populations across Colombia and México has revealed distinct eco-geographical clades with varying behaviours, habitat preferences, and vectorial capacities.[6,7,8,41] Although such population structure remains uncharacterised in Panamá, the high infection rate and peridomestic adaptation observed in Palmira Arriba suggest the possible existence of a behaviourally and genetically distinct ecotype. Future molecular analyses, such as COI barcoding, ITS-2 genotyping, or microsatellite profiling, could help delineate population boundaries and guide the design of vector control strategies adapted to local ecological and epidemiological conditions.[11,27,28]

Taken together, these findings highlight *T. dimidiata* as an important and ecologically adaptable vector in western Panamá, capable of sustaining *T. cruzi* transmission in peridomestic environments at high altitude. The detection of TcIDOM and TcIa genotypes reinforces the epidemiological relevance of this species in domestic and sylvatic cycles. However, this pilot study, while providing the first molecular confirmation of *T. cruzi* infection in *T. dimidiata* from Palmira Arriba, is constrained by its relatively small sample size, limited geographic scope, and the resolution of the molecular tools used for parasite genotyping. Parasite load was not quantified in

infected specimens, which could have offered further insights into vector competence and intra-vector parasite dynamics. The absence of mixed infections or additional DTUs (*e.g.*, TcId, TcIV) may result from methodological limitations or restricted sampling depth rather than their true absence in the local transmission network. Likewise, blood meal analysis was limited to conventional PCR targeting the cyt b gene, which, although informative, lacks the sensitivity and specificity to detect and characterise low-abundance host DNA.

To address these limitations and obtain a more comprehensive understanding of *T. cruzi* transmission dynamics in the region, future research should incorporate large scale and longitudinal entomological surveys, expanded ecological sampling across seasons and microhabitats, and the application of high-resolution molecular tools such as next-generation sequencing (NGS), multilocus sequence typing (MLST), and microsatellite analysis. In parallel, it is necessary to assess *T. cruzi* infection in both domestic and sylvatic mammalian reservoirs, as well as in human populations, to characterise the complete transmission network and identify potential spillover events. These integrative approaches will be essential to reveal cryptic parasite diversity, evaluate population structure in *T. dimidiata*, and determine whether a distinct behavioural or genetic ecotype is emerging in the highlands of Chiriquí. Finally, these findings will support the development of more effective One Health-based surveillance and vector control strategies for this ecologically complex and understudied region, approaches that have already proven successful in Central American countries.[42]

## ACKNOWLEDGEMENTS

To Mr José Montenegro and Mr Issac Montenegro for their participation in triatomine collection and fieldwork activities. We also thank the staff of the Vector Control Department of the Ministry of Health (MINSA) in Chiriquí for their invaluable guidance and logistical support in coordinating collection sites. Special thanks are extended to the residents of the Palmira Arriba community for their collaboration and support throughout the study.

## AUTHORS' CONTRIBUTION

VJP - investigation, methodology, writing-review & editing, data curation; KG - methodology, investigation, writing-review & editing; JEC - conceptualisation, investigation, writing-review & editing; AS - conceptualisation, investigation, writing-original draft, writing-review & editing. The authors declare that they have no known competing financial interests or personal relationships that could have appeared to influence the work reported in this paper.

## DATA AVAILABILITY

The contents underlying the research text are included in the manuscript.

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

# OPEN PEER REVIEW

Memórias do IOC thanks the anonymous reviewers for their contribution to the peer review of this work.

**FIRST REVIEW ROUND**

REVIEWERS' COMMENTS

**REVIEWER #1**

I found the manuscript "Molecular Characterization of Trypanosoma cruzi in Triatoma dimidiata from a Highland Locality in Western Panamá" very interesting", although the capture n is modest, the authors made an effort to identify several aspects of the vector-parasite-reservoir relationship, however, there are some aspects that should be addressed to improve the manuscript.

The paragraph "In support of ongoing surveillance efforts, the Department of Parasitology Research at the Gorgas Memorial Institute has received 30 triatomine specimens from Palmira Arriba over the past decade. These submissions were processed as part of a diagnostic service focused on species identification and assessment of T. cruzi infection status. All specimens were taxonomically confirmed as T. dimidiata and collected from peridomestic ecotopes. Microscopic examination revealed that 16.7% (5/30) of the triatomines were infected with T. cruzi." It leaves me with many questions. One of them is, who did this work? Has it been published? Is it the same area they're trying to study, or is there a difference?

The following paragraph corresponds to the study area in materials and methods: "Palmira Arriba, located at approximately 1,300 meters above sea level in the Boquete highlands, constitutes an ecologically distinct area that, until now, had not been the subject of molecular studies concerning triatomine-borne T. cruzi transmission." However, it is necessary to know why this area was chosen and to include the questions: How many studies exist, how they were conducted, and what differences do they propose? Is there an impact on people? Are there studies on the presence of T. cruzi in residents?

The following paragraph, like the previous ones, has no references; it is important to attach the source from which the data was obtained.:

"Palmira Arriba (8.76716° N, 82.45815° W) is a small rural community located in the highlands of Boquete District, Chiriquí Province, Panamá, approximately 478 kilometers (a 7-hour drive) from Panamá City (Figure 1). The area has an average…." Ref?

In the following paragraph: "Taxonomic identification was performed to confirm species and developmental stage. What keys did they use and as far as I know, there are no keys for nymphs, very poor for 5th instar but they don't exist for the others?

Could you explain the paragraph? "Five peridomestic areas in Palmira Arriba were surveyed.." What areas are they? Why didn't they specify them in the Materials and Methods section? How far apart are they from each other? Or were they referring to houses?

All specimens were taxonomically confirmed as T. dimidiate". Once again, there are no taxonomic keys for nymphs.

In the following paragraph: "Genotyping revealed that 76.2% (16/21) of the positive samples corresponded to the TcI lineage, predominantly the TcIDOM genotype and TcIa haplotype."  Should they add the statement that the remaining sequences could not be genotyped? And I'm left wondering why it couldn't be done?

In the following paragraph: "The analysis of blood meal sources using Cyt b primers revealed that one triatomines tested positive for mammalian blood and two for avian blood." Why were feeding habits identified in only 3 insects?

In the following paragraph: "This study provides the first molecular confirmation of T. cruzi infection…" In the introduction they make you think that there is a previous study, that is correct, so this paragraph would not be true.

This line under discussion "Located at approximately 1,300 meters above sea level.." It has already been mentioned 4 times throughout the manuscript, so it is no longer necessary to do so again.

AUTHORS' RESPONSE TO THE REVIEWERS

Panamá, October 14, 2025
Dr. Adeilton Brandão
Handling Editor
Memórias do Instituto Oswaldo Cruz

We sincerely thank Reviewer 1 for the thoughtful and constructive comments that have helped us to improve the clarity, rigor, and contextualization of our manuscript. All comments were carefully considered, and corresponding modifications were made in the revised version of the manuscript entitled "Molecular

Characterization of Trypanosoma cruzi in Triatoma dimidiata from a Highland Locality in Western Panamá." Below we provide a point-by-point response to each observation. Reviewer comments are presented in italics, followed by our detailed responses.

1. "The paragraph about 30 triatomine specimens received at Gorgas leaves many questions. Who did this work? Has it been published? Is it the same area, or is there a difference?"

Response: We appreciate this observation. The Introduction was revised to clarify that these specimens were processed by the Department of Parasitology Research at the Gorgas Memorial Institute as part of a diagnostic service, not as a published research study. We also explicitly indicated that the samples originated from the same community of Palmira Arriba but were independent of the present field collections. This clarification prevents confusion regarding data origin. Furthermore, the recent and recurring reception of triatomine specimens infected with Trypanosoma cruzi from this same community at our department raised a public health alert, expressed both by community residents and by personnel from the Ministry of Health's Vector Control Department. This situation prompted the decision to conduct entomological surveillance activities in the area.

2. "It is necessary to know why this area was chosen and to include how many studies exist, how they were conducted, and what differences they propose. Is there an impact on people? Are there studies on residents?"

Response: We agree and have expanded the Introduction to justify the selection of Palmira Arriba. The text now cites historical entomological surveys conducted in 1986, 1999, and 2012 (Tason de Camargo, 2020), and mentions preliminary serological screening of local residents that showed 2% positivity for T. cruzi antibodies. We also emphasized that no molecular investigations had been previously conducted in this locality. This background underscores the novelty and relevance of our study.

3. "The following paragraph has no references; it is important to attach the source from which the data was obtained."

Response: We have added references to the "Study Site" section indicating that geographical and demographic information was obtained from the National Geographic Institute "Tommy Guardia." (2023); Empresa de Transmisión Eléctrica S.A. (ETESA). (2023). Meteorological data from the Hydrometeorology Directorate of Panamá; National Institute of Statistics and Census (INEC). (2023). Agricultural statistics of the Republic of Panamá and Ministry of Environment (MiAmbiente). (2023). Forest cover and land use of the Republic of Panamá 2023. These additions strengthen the reliability of the contextual data.

4. "Taxonomic identification was performed... What keys did they use? There are no keys for nymphs."

Response: This has been clarified in the "Methods" section. Adult specimens were identified following the morphological keys of Lent & Wygodzinsky (1979) and Méndez & Sousa (1979). For nymphal stages, identification was based on rearing representative individuals to adulthood in the laboratory and corroborating distinctive morphological traits observable in late instars.

We have acknowledged this as a potential limitation in nymphal identification.

5. "Could you explain the paragraph 'Five peridomestic areas in Palmira Arriba were surveyed…'? What areas are they? Why didn't they specify them? How far apart are they?"

Response: We have revised the corresponding section in the "Methods" to specify that five distinct household peridomestic areas were surveyed, located within 0.5-1.5 km of each other. These included piles of timber and wooden materials, animal enclosures (dog/cat resting areas and chicken coops), and adjacent storage structures within household premises. This addition provides a clearer description of the sampling design.

6. "Genotyping revealed that 76.2%… Should they add that the remaining sequences could not be genotyped? Why?"

Response: We thank the reviewer for this important suggestion. The Results section now includes a statement indicating that 23.8% of the positive samples could not be genotyped, likely due to low DNA concentration or degraded templates, a limitation often encountered in field-collected triatomines. This information was also mentioned in the Discussion to acknowledge its relevance.

7. "Why were feeding habits identified in only 3 insects?"

Response: This is an important observation raised by the reviewer. We have clarified in the Results that most specimens were starved and lacked recent blood meals, limiting the PCR detection of vertebrate DNA. A citation to Moo-Millan et al. (2019) was added to support this explanation and to indicate that such limitations are common in field studies.

8. "This study provides the first molecular confirmation... In the Introduction they make you think there is a previous study, so this would not be true."

Response: We appreciate this important observation. The statement was modified to read: "This study provides the first molecular confirmation of T. cruzi infection in T. dimidiata from Palmira Arriba, a highland rural locality in Chiriquí Province." This change avoids any ambiguity regarding previous research and accurately reflects the novelty of our findings.

9. "The line 'Located at approximately 1,300 meters above sea level…' is repeated four times."

Response: We agree with this comment. Redundant mentions of altitude have been removed throughout the manuscript, keeping only two necessary references, one in the Study Site description and one in the Discussion for ecological comparison.

We sincerely thank Reviewer 1 once again for the valuable comments and suggestions, which have significantly improved the clarity, methodological detail, and scientific value of our manuscript. We hope that the revised version meets the expectations of the reviewers and the editorial board.

Sincerely yours,

Dr. Azael Saldaña Patiño
Centro de Investigación y Diagnóstico de Enfermedades Parasitarias (CIDEP),
Facultad de Medicina, Universidad de Panamá, Panamá
E-mail: azael.saldana@up.ac.pa

Dr. José E. Calzada
Departamento de Investigaciones en Parasitología,
Instituto Conmemorativo Gorgas de Estudios de la Salud (ICGES), Panamá
E-mail: jecalzada@gorgas.gob.pa
(On behalf of all co-authors: Vanessa Pineda and Kadir González)

## SECOND REVIEW ROUND

REVIEWERS' COMMENTS

### REVIEWER #1

Dear editor, reviewing the authors' responses. Pineda, Gonzáleza, Calzadaa, Saldaña, and his manuscript entitled "Molecular Characterization of Trypanosoma cruzi in Triatoma dimidiata from a Highland Locality in Western Panamá" I find your answers adequate, so I have no further comments on your manuscript.

