## [Reviewer Report · FIRST REVIEW ROUND - REVIEWERS COMMENTS]

## REVIEWER #1

I found the manuscript *“Molecular Characterization of Trypanosoma cruzi in Triatoma dimidiata from a Highland Locality in Western Panamá”* very interesting”, although the capture n is modest, the authors made an effort to identify several aspects of the vector-parasite-reservoir relationship, however, there are some aspects that should be addressed to improve the manuscript.

The paragraph “In support of ongoing surveillance efforts, the Department of Parasitology Research at the Gorgas Memorial Institute has received 30 triatomine specimens from Palmira Arriba over the past decade. These submissions were processed as part of a diagnostic service focused on species identification and assessment of T. cruzi infection status. All specimens were taxonomically confirmed as T. dimidiata and collected from peridomestic ecotopes. Microscopic examination revealed that 16.7% (5/30) of the triatomines were infected with T. cruzi.”

It leaves me with many questions. One of them is, who did this work? Has it been published?

Is it the same area they’re trying to study, or is there a difference?

The following paragraph corresponds to the study area in materials and methods: “Palmira Arriba, located at approximately 1,300 meters above sea level in the Boquete highlands, constitutes an ecologically distinct area that, until now, had not been the subject of molecular studies concerning triatomine-borne T. cruzi transmission.”

However, it is necessary to know why this area was chosen and to include the questions: How many studies exist, how they were conducted, and what differences do they propose?

Is there an impact on people? Are there studies on the presence of T. cruzi in residents?

The following paragraph, like the previous ones, has no references;

it is important to attach the source from which the data was obtained.:

“Palmira Arriba (8.76716° N, 82.45815° W) is a small rural community located in the highlands of Boquete District, Chiriquí Province, Panamá, approximately 478 kilometers (a 7-hour drive) from Panamá City (Figure 1). The area has an average….”

Ref?

In the following paragraph: “Taxonomic identification was performed to confirm species and developmental stage. What keys did they use and as far as I know, there are no keys for nymphs, very poor for 5th instar but they don’t exist for the others?

Could you explain the paragraph? “Five peridomestic areas in Palmira Arriba were surveyed..” What areas are they?

Why didn’t they specify them in the Materials and Methods section? How far apart are they from each other?

Or were they referring to houses?

All specimens were taxonomically confirmed as T. dimidiate”. Once again, there are no taxonomic keys for nymphs.

In the following paragraph: “Genotyping revealed that 76.2% (16/21) of the positive samples corresponded to the TcI lineage, predominantly the TcIDOM genotype and TcIa haplotype.”

Should they add the statement that the remaining sequences could not be genotyped?

And I’m left wondering why it couldn’t be done?

In the following paragraph: “The analysis of blood meal sources using Cyt b primers revealed that one triatomines tested positive for mammalian blood and two for avian blood.”

Why were feeding habits identified in only 3 insects?

In the following paragraph: “This study provides the first molecular confirmation of T. cruzi infection…” In the introduction they make you think that there is a previous study, that is correct, so this paragraph would not be true.

This line under discussion “Located at approximately 1,300 meters above sea level..” It has already been mentioned 4 times throughout the manuscript, so it is no longer necessary to do so again.

## AUTHORS’ RESPONSE TO THE REVIEWERS

Panamá, October 14, 2025

Dr. Adeilton Brandão

Handling Editor

Memórias do Instituto Oswaldo Cruz

We sincerely thank Reviewer 1 for the thoughtful and constructive comments that have helped us to improve the clarity, rigor, and contextualization of our manuscript.

All comments were carefully considered, and corresponding modifications were made in the revised version of the manuscript entitled “Molecular Characterization of Trypanosoma cruzi in Triatoma dimidiata from a Highland Locality in Western Panamá.”

Below we provide a point-by-point response to each observation. Reviewer comments are presented in italics, followed by our detailed responses.

1. *“The paragraph about 30 triatomine specimens received at Gorgas leaves many questions. Who did this work? Has it been published? Is it the same area, or is there a difference?”*

**Response:** We appreciate this observation. The Introduction was revised to clarify that these specimens were processed by the Department of Parasitology Research at the Gorgas Memorial Institute as part of a diagnostic service, not as a published research study.

We also explicitly indicated that the samples originated from the same community of Palmira Arriba but were independent of the present field collections.

This clarification prevents confusion regarding data origin. Furthermore, the recent and recurring reception of triatomine specimens infected with Trypanosoma cruzi from this same community at our department raised a public health alert, expressed both by community residents and by personnel from the Ministry of Health’s Vector Control Department.

This situation prompted the decision to conduct entomological surveillance activities in the area.

2. *“It is necessary to know why this area was chosen and to include how many studies exist, how they were conducted, and what differences they propose. Is there an impact on people? Are there studies on residents?”*

**Response:** We agree and have expanded the Introduction to justify the selection of Palmira Arriba.

The text now cites historical entomological surveys conducted in 1986, 1999, and 2012 (Tason de Camargo, 2020), and mentions preliminary serological screening of local residents that showed 2% positivity for T. cruzi antibodies.

We also emphasized that no molecular investigations had been previously conducted in this locality.

This background underscores the novelty and relevance of our study.

3. *“The following paragraph has no references; it is important to attach the source from which the data was obtained.”*

**Response:** We have added references to the “Study Site” section indicating that geographical and demographic information was obtained from the National Geographic Institute “Tommy Guardia.”

(2023); Empresa de Transmisión Eléctrica S.A. (ETESA). (2023). Meteorological data from the Hydrometeorology Directorate of Panamá;

National Institute of Statistics and Census (INEC). (2023). Agricultural statistics of the Republic of Panamá and Ministry of Environment (MiAmbiente).

(2023). Forest cover and land use of the Republic of Panamá 2023. These additions strengthen the reliability of the contextual data.

4. *“Taxonomic identification was performed... What keys did they use? There are no keys for nymphs.”*

**Response:** This has been clarified in the “Methods” section. Adult specimens were identified following the morphological keys of Lent & Wygodzinsky (1979) and Méndez & Sousa (1979).

For nymphal stages, identification was based on rearing representative individuals to adulthood in the laboratory and corroborating distinctive morphological traits observable in late instars.

We have acknowledged this as a potential limitation in nymphal identification.

5. *“Could you explain the paragraph ‘Five peridomestic areas in Palmira Arriba were surveyed…’? What areas are they? Why didn’t they specify them? How far apart are they?”*

**Response:** We have revised the corresponding section in the “Methods” to specify that five distinct household peridomestic areas were surveyed, located within 0.5-1.5 km of each other.

These included piles of timber and wooden materials, animal enclosures (dog/cat resting areas and chicken coops), and adjacent storage structures within household premises.

This addition provides a clearer description of the sampling design.

6. *“Genotyping revealed that 76.2%… Should they add that the remaining sequences could not be genotyped? Why?”*

**Response:** We thank the reviewer for this important suggestion. The Results section now includes a statement indicating that 23.8% of the positive samples could not be genotyped, likely due to low DNA concentration or degraded templates, a limitation often encountered in field-collected triatomines.

This information was also mentioned in the Discussion to acknowledge its relevance.

7. *“Why were feeding habits identified in only 3 insects?”*

**Response:** This is an important observation raised by the reviewer.

We have clarified in the Results that most specimens were starved and lacked recent blood meals, limiting the PCR detection of vertebrate DNA.

A citation to Moo-Millan et al. (2019) was added to support this explanation and to indicate that such limitations are common in field studies.

8. *“This study provides the first molecular confirmation... In the Introduction they make you think there is a previous study, so this would not be true.”*

**Response:** We appreciate this important observation. The statement was modified to read: “This study provides the first molecular confirmation of T. cruzi infection in T. dimidiata from Palmira Arriba, a highland rural locality in Chiriquí Province.”

This change avoids any ambiguity regarding previous research and accurately reflects the novelty of our findings.

9. *“The line ‘Located at approximately 1,300 meters above sea level…’ is repeated four times.”*

**Response:** We agree with this comment. Redundant mentions of altitude have been removed throughout the manuscript, keeping only two necessary references, one in the Study Site description and one in the Discussion for ecological comparison.

We sincerely thank Reviewer 1 once again for the valuable comments and suggestions, which have significantly improved the clarity, methodological detail, and scientific value of our manuscript.

We hope that the revised version meets the expectations of the reviewers and the editorial board.

Sincerely yours,

Dr. Azael Saldaña Patiño

Centro de Investigación y Diagnóstico de Enfermedades Parasitarias (CIDEP),

Facultad de Medicina, Universidad de Panamá, Panamá

E-mail: azael.saldana@up.ac.pa

Dr. José E. Calzada

Departamento de Investigaciones en Parasitología,

Instituto Conmemorativo Gorgas de Estudios de la Salud (ICGES), Panamá

E-mail: jecalzada@gorgas.gob.pa

(On behalf of all co-authors: Vanessa Pineda and Kadir González)

---

## [Reviewer Report · REVIEWERS COMMENTS]

## REVIEWER #1

Dear editor, reviewing the authors’ responses. Pineda, Gonzáleza, Calzadaa, Saldaña, and his manuscript entitled “Molecular Characterization of Trypanosoma cruzi in Triatoma dimidiata from a Highland Locality in Western Panamá” I find your answers adequate, so I have no further comments on your manuscript.